# Long-Term Longitudinal Analysis of Neutralizing Antibody Response to Three Vaccine Doses in a Real-Life Setting of Previously SARS-CoV-2 Infected Healthcare Workers: A Model for Predicting Response to Further Vaccine Doses

**DOI:** 10.3390/vaccines10081237

**Published:** 2022-08-02

**Authors:** Saverio Giuseppe Parisi, Carlo Mengoli, Monica Basso, Ilaria Vicenti, Francesca Gatti, Renzo Scaggiante, Lia Fiaschi, Federica Giammarino, Marco Iannetta, Vincenzo Malagnino, Daniela Zago, Filippo Dragoni, Maurizio Zazzi

**Affiliations:** 1Department of Molecular Medicine, University of Padova, Via Gabelli, 63, 35100 Padova, Italy; mengolicarlo@gmail.com (C.M.); monica.basso@unipd.it (M.B.); francesca.gatti@asst-garda.it (F.G.); daniela.zago.3@studenti.unipd.it (D.Z.); 2Department of Medical Biotechnologies, University of Siena, Viale Bracci 16, 53100 Siena, Italy; vicenti@unisi.it (I.V.); lia300790@gmail.com (L.F.); federica.giammari@gmail.com (F.G.); dragoni16@student.unisi.it (F.D.); maurizio.zazzi@unisi.it (M.Z.); 3Belluno Hospital, Viale Europa, 22, 32100 Belluno, Italy; renzo.scaggiante@aulss1.veneto.it; 4Infectious Disease Unit, Department of System Medicine, Tor Vergata University and Hospital, Via Montpellier 1, 00133 Rome, Italy; marco.iannetta@uniroma2.it (M.I.); malagninovincenzo@gmail.com (V.M.); 5Department of Medicine, University of Udine, Via Colugna 50, 33100 Udine, Italy

**Keywords:** COVID-19, long-term follow-up, third vaccine dose, BNT162b2 mRNA vaccine, authentic virus neutralization, healthcare workers, mild or asymptomatic SARS-CoV-2 infection, no reinfection

## Abstract

We report the time course of neutralizing antibody (NtAb) response, as measured by authentic virus neutralization, in healthcare workers (HCWs) with a mild or asymptomatic SARS-CoV-2 (severe acute respiratory syndrome coronavirus 2) infection diagnosed at the onset of the pandemic, with no reinfection throughout and after a three-dose schedule of the BNT162b2 mRNA vaccine with an overall follow-up of almost two years since infection. Forty-eight HCWs (median age 47 years, all immunocompetent) were evaluated: 29 (60.4%) were asymptomatic. NtAb serum was titrated at eight subsequent time points: T1 and T2 were after natural infection, T3 on the day of the first vaccine dose, T4 on the day of the second dose, T5, T6, and T7 were between the second and third dose, and T8 followed the third dose by a median of 34 days. NtAb titers at all postvaccination time points (T4 to T8) were significantly higher than all those at prevaccination time points (T1 to T3). The highest NtAb increase was following the first vaccine dose while subsequent doses did not further boost NtAb titers. However, the third vaccine dose appeared to revive waning immunity. NtAb levels were positively correlated at most time points suggesting an important role for immunogenetics.

## 1. Introduction

The coronavirus disease 2019 (COVID-19) pandemic persists and 2022 is the third year with the disease as a worldwide major public health problem. Numerous genetically distinct lineages have evolved since the emergence of the original Wuhan Hu-1 strain of severe acute respiratory syndrome coronavirus 2 (SARS-CoV-2). Among these, the Alpha (B.1.1.7), Delta (B.1.617.2), and Omicron (B.1.1.529) variants were responsible for new waves of infection [1,2,3,4] due to increased transmissibility.

First approved in December 2020 [5], COVID-19 vaccines remain the cornerstone of prevention and protection against infection and severe disease. While the initial one- or two-dose schedule, depending on the vaccine, has played a key role in the mitigation of COVID-19 morbidity and mortality, the emergence of viral variants with varying degrees of immune escape led most countries to deploy a third dose or even a fourth dose of vaccine boosters [6,7,8,9]. Overall, the interplay between natural infection and vaccination, as well as the role of different vaccine schedules and methods used to quantify the neutralizing antibody (NtAb) response [10,11,12,13,14], have made it difficult to depict the key features and dynamics of immunization to SARS-CoV-2 in a real-life setting.

Here, we report the time course of NtAb response, as measured by authentic virus neutralization, in healthcare workers (HCWs) with a mild or asymptomatic SARS-CoV-2 infection diagnosed at the onset of pandemic and no reinfection throughout and after a three-dose schedule of BNT162b2 mRNA vaccine with an overall follow-up of almost two years since infection.

## 2. Materials and Methods

### 2.1. Study Design

Forty-eight HCWs living in Northern Italy were included in the study. All of them had a laboratory diagnosis of SARS-CoV-2 infection in the Veneto region in March–April 2020 and were tested because of clinical suspicion or in the context of the hospital surveillance program. Symptomatic HCWs were evaluated by an infectious disease specialist and diagnosed with mild disease [15], as defined by the symptoms reported in Appendix A. Based on a hospital nasopharyngeal swab screening surveillance program performed at different intervals (ranging from four to seven days, according to the epidemiological context), no reinfection was detected during the whole study period. After a median interval of 297 days from the diagnosis of their SARS-CoV-2 infection, the HCWs received the first dose of the BNT162b2 vaccine, followed by the second dose after three weeks, and then by the third dose 9–12 months later. Written informed consent was obtained from all the HCWs willing to participate in a prospective study of the virus’ NtAb response, which was approved by the Comitato per la Sperimentazione Clinica di Treviso e Belluno (prot 812/2020) and performed in accordance with the ethical standards as laid down in the Declaration of Helsinki. NtAb serum was titrated at eight subsequent time points (T1 to T8) (Figure 1).

### 2.2. Titration of Virus Neutralizing Antibodies

NtAbs to the live B.1 lineage virus (GISAID accession number EPI_ISL_2472896) were titrated in duplicate by testing two-fold serial dilutions of sera, starting at 1/10, with 100 TCID_50_ of the virus in VERO E6 cells in a 96-well plate. Virus-induced cell death was calculated by automated measurement of cell viability by the Cell-titer Glo 2.0 system in a GloMax Discover luciferase plate reader (Promega, Madison, WI, USA) [16]. The NtAb titer (ID_50_) was defined as the reciprocal value of the sample dilution that showed 50% protection from the virus-induced cytopathic effect. Each run included an uninfected cell control, an infected cell control, and the virus back titration to confirm the virus inoculum.

### 2.3. Statistical Analysis

Antibody levels at the eight time points were reported as median, 25th, and 75th percentiles, minima (the lowest value detected), and maxima (the highest value detected). Sera with ID_50_ < 10 were defined as negative and scored as 5 for statistical analysis. Since the NtAb value distribution was strongly right-skewed at all time points, the symmetry of the distribution was ameliorated by log_10_-transformation. Clinical symptoms were categorized as absent or mild. The data structure was explored by calculating the pairwise correlation structure of the following variables: T1 to T8, age (expressed as years absolute value), gender (male or female), and symptomatic infection. The Spearman and Wilcoxon test (null hypothesis is equal medians) rank methods were used, as appropriate.

The effects of the sequential series of time points, age, gender, and symptomatic infection on the antibody levels were assessed by performing a mixed-model linear regression, where the dependent variable was log_10_-transformed antibody titer, and the predictors were time points, gender = female, symptomatic infection, and age. All predictors, except age, were categorical. The variable “time points” comprised eight levels. The original antibody titers had a right-skewed distribution at each time point. For this reason, a logarithmic transformation was performed; the ensuing distribution was symmetric and Gaussian (normal) with reasonable approximation. This permitted the use of the mixed-model linear regression approach. Notably, the logarithmic transformation of antibody titers is a common practice [17].

The use of the mixed-model linear regression where time, the main predictor, was a multinomial categorical variable (time points), instead of a continuous variable, was chosen because we felt that the linear model was too rigid in order to explain the multiphasic evolution of the antibody titer. The mixed-model linear regression is preferable when there are repeated measures in the same subjects. This model is fit to correctly interpret the role of different subjects, obtaining the best available evaluation of the mean titer, and the best confidence intervals, at each time point.

## 3. Results

Forty-eight HCWs (31 females and 17 males) were evaluated: the median age was 47 (IQR 40–53) years and 29 (60.4%) were asymptomatic. All HCWs were immunocompetent and comorbidity was present in six of them (four presented with uncomplicated arterial hypertension and two with dyslipidemia).

NtAbs were undetectable in two out of 39 HCWs (5.1%) at T1, six out of 38 (15.8%) at T2, and three out of 31 (9.7%) at T3 but in none of the subjects following vaccination. Figure 2 shows the log_10_-transformed antibody levels at the eight time points. One patient (female, 57 years, mild disease) had a left outlier NtAb titer at T4 (1.0 log) and a right outlier at T5 (4.18 log), and two patients (female, 33 years, asymptomatic and female, 64 years, asymptomatic) had left outliers at T6 (1.8 and 1.9 log, respectively).

NtAb titers from all postvaccination time points (T4 to T8) were significantly higher than those from all prevaccination time points (T1 to T3) (*p* < 0.0001). The highest increase with respect to the previous time point was detected at T4, i.e., three weeks following the first vaccine dose. None of the subsequent vaccine doses triggered NtAb titers significantly higher than T4, however, the third vaccine dose significantly revived waning NtAb titers (*p* < 0.0001 for comparison between T8 and T7).

Pairwise correlations between log_10_-transformed NtAb titers measured at different time points are shown in Appendix A. NtAb levels were positively correlated at most time points and symptomatic infection was positively correlated to antibody levels at several time points as well. The coefficients of the mixed-model linear regression are reported in Table 1. Age predicted lower NtAb levels, whereas symptomatic infection predicted higher NtAb levels significantly.

Neutralizing antibody titers in asymptomatic and symptomatic HCWs as predictive margins after mixed-model linear regression are depicted in Figure 3.

## 4. Discussion

Most studies assessing the efficacy of the third vaccine dose have included patients with no previous COVID-19 infection or with unknown SARS-CoV-2 infection status or have focused on patients with immunosuppression [18]. Our real-life analysis aimed to describe the long-term dynamics of NtAb titers in a cohort of HCWs with asymptomatic or mild wild-type SARS-CoV-2 infection followed by a three-dose vaccine schedule and no reinfection up to more than 600 days.

The NtAb titer after a median of 34 days from the third vaccine dose was significantly higher with respect to the preceding time point available (7 months after the second dose and approximately three months before the third dose), implying that a third antigenic stimulation can raise a waning NtAb response in subjects undergoing a complete vaccination cycle following natural infection. However, NtAb levels measured around 3–4 weeks following the first, second, and third vaccine doses did not differ significantly from one another. This suggests that, in subjects previously experiencing mild or asymptomatic infection, subsequent vaccine shots do not boost humoral immunity to higher-than-ever levels but rather refresh the waning immune system to comparable levels. However, in another HCW cohort with a median age of 41 years, Romero-Ibarguengoitia et al. [19] reported a significant increase in IgG titers detected 21–28 days after the third dose with respect to the IgG value at 21–28 days after the second dose. Potentially relevant differences with respect to our study include the use of a commercial anti-spike IgG assay instead of authentic neutralization, the inclusion of a proportion of subjects infected twice, the shorter interval (6 months versus more than nine months) between second and third vaccine dose and possibly disease severity (not reported). Omicron spreading caused the ongoing wave of the COVID-19 epidemic: this variant is characterized by more than 30 amino acid substitutions in the S protein and these changes cover almost all of the key mutations of Alpha, Beta, Gamma, and Delta VOC [20]. Since November 2021, when it was identified, Omicron continuously evolved: BA.2.12.1, BA.4, BA.5, BA.2.9.1, BA.2.13, and BA.2.11 are the Omicron lineages first detected from December 2021 to March 2022 included in the World Health Organization (WHO) variants of concern lineages under monitoring as of the beginning of June 2022 [21]. This rapidly evolving scenario had a strong impact on public health: the Omicron variants of SARS-CoV-2 have greater transmissibility than the previously identified variants [22,23]. It is important to note that a third dose of vaccine has been shown to partially restore the neutralization activity against the highly divergent Omicron variant in subjects not previously infected by SARS-CoV-2 [24] and in those with a previous infection [25,26]. In addition, a positive impact on antibody response was demonstrated for high preinfection antibody titers [27]. Nevertheless, recent data showed that omicron variants can evade the humoral immune response following the booster dose with the BNT162b2 vaccine (with a reduction in neutralizing antibody titers ranging from a factor of 6.4 for BA.1 to a factor of 21.0 for BA.4 or BA.5. with respect to reference WA1/2020 isolate) and subvariants BA.2.12.1, BA.4, and BA.5 escape neutralizing responses against a previous BA.1 or BA.2 infection [28]. Further, Omicron may evolve mutations to evade the humoral immunity elicited by a BA.1 infection, suggesting that BA.1-derived vaccine boosters may not achieve broad-spectrum protection against new Omicron variants [29].

Goldblatt et al. [30] estimated a protective threshold in vaccine recipients corresponding to 154 binding antibody units /mL to the original viral strain by using a population-based model. Again, the titration method used was an anti-spike IgG binding assay and it can be difficult to correlate these data with live virus NtAb titers. In our dataset, the lowest median postvaccination NtAb titer was 421 ID_50_ at 9–10 months before the administration of the third dose (T7) and this titer likely continued to decrease to a minimum just before this third dose recall. However, no reinfection was demonstrated despite the very high transmissibility of the Delta [31] and Omicron variants [32] which were prevalent in the last seven and two months of observation, respectively.

The availability of multiple time points data for each HCW allowed us to describe the dynamics of the NtAb response from immunity to natural infection across multiple vaccine stimulations. Overall, NtAb titers were correlated at the different time points, confirming the result published by Mantus et al. [33], who observed a positive correlation between antireceptor binding domain (RBD) antibodies, anti-spike IgG, NtAb titers, and RBD+ memory B cells prior to and after vaccination in subjects recovering from infection. The correlation between natural and artificial active immunity suggests an important role for immunogenetics, possibly involving differences in the individual B cell receptor repertoire [34] and HLA haplotype [35].

The strengths of the study include the prospective design encompassing almost two years with a study population regularly monitored to rule out breakthrough infections and the use of an authentic live virus neutralization assay; the main limitations are the low sample size, the incomplete availability of samples at intermediate time points, and no evaluation of the cross-protection against the emerging variants. However, the kinetics of the response to the ancestral virus provides useful information on the durability of the immunological memory and on the ability to respond to further stimuli.

## 5. Conclusions

We will continue to study the NtAb response over time after the third dose and immediately before and after the fourth dose that is now recommended in Italy for people over sixty and may be recommended for HCW too. This will be continued with the aim to evaluate over time the changes in titer against the ancestral virus, describe titer against Omicron subvariants and, at the same time, monitor the occurrence of COVID-19 reinfection in this high-risk cohort in order to describe the characteristic of humoral response after three or four vaccine stimulations and one or two natural infections.

In addition to being able to pursue new variants and perhaps responses to new vaccine antigens, we have, however, so far been able to study the net kinetics of response to repeated antigenic stimuli that were temporally and quantitatively well documented against the new coronavirus.

We have thus documented the effects of up to four stimuli (natural infection and vaccine boosters) over time, demonstrating similar and non-decreasing titers. This is relevant in terms of public health. We encourage scientists to design studies allowing prolonged regular follow-up of specific categories to define how infection and vaccination determine the durability of protective immunity in the context of the ongoing epidemic.

## Figures and Tables

**Figure 1 vaccines-10-01237-f001:**
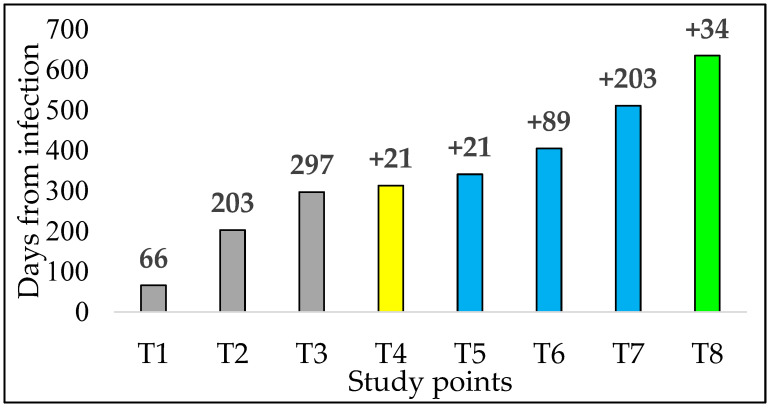
Sampling timeline following the diagnosis of SARS-CoV-2 infection. The gray bars correspond to study points in not yet vaccinated healthcare workers, the yellow bar to the study point after the first vaccine dose, the light blue bars to three different study points after the second vaccine dose, and the green bar to the study point after the third dose. Data for the columns of the histogram are expressed as the median value of days from diagnosis to NtAb testing at each study point: intervals from infection (grey) or vaccinations (yellow, blue, and green) are indicated on the top. Days after the vaccine dose are expressed as the median and interquartile range (IQR). T1, after natural infection. T2: after natural infection. T3: after natural infection, day of the first vaccine dose. T4: 21 days after the first vaccine dose, (median [IQR 20–21]), day of the second vaccine dose. T5: 21 days after the second vaccine dose (median [IQR 20–23]). T6: 89 days after the second vaccine dose (median [IQR 86–98]). T7: 203 days after the second vaccine dose (median [IQR 183–220]). T8: 34 days after the third vaccine dose (median [IQR 30–44]) administered 285 days after the second (median [IQR 273–302]).

**Figure 2 vaccines-10-01237-f002:**
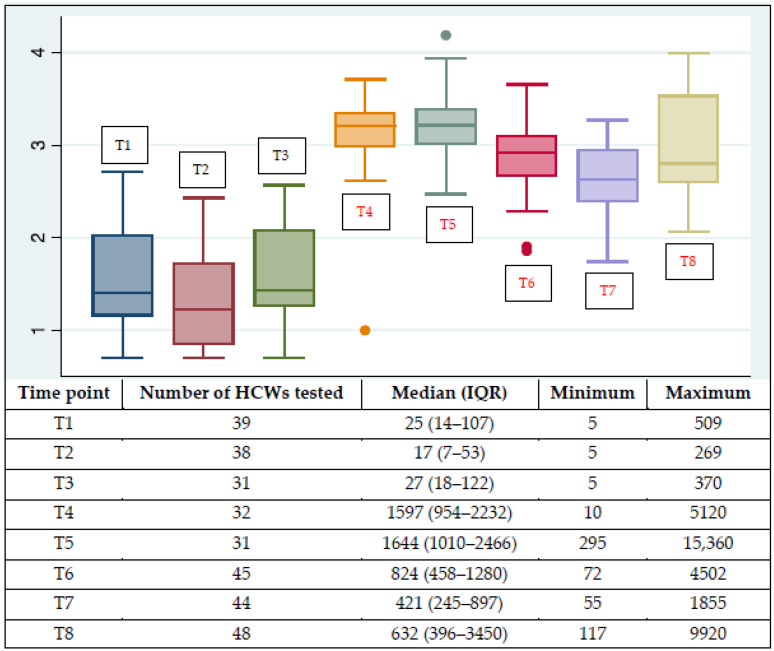
Log_10_-transformed antibody levels (T1–T8) at the eight time points considered. Medians with the interquartile interval and range, along with some outliers, are shown in the figure. Postvaccine study points are denoted by the red font color. The table under the figure shows absolute values. NtAb titers were expressed as ID_50_ (reciprocal value of the sample dilution that showed 50% protection of virus cytopathic effect). HCWs: healthcare workers. IQR: interquartile range.

**Figure 3 vaccines-10-01237-f003:**
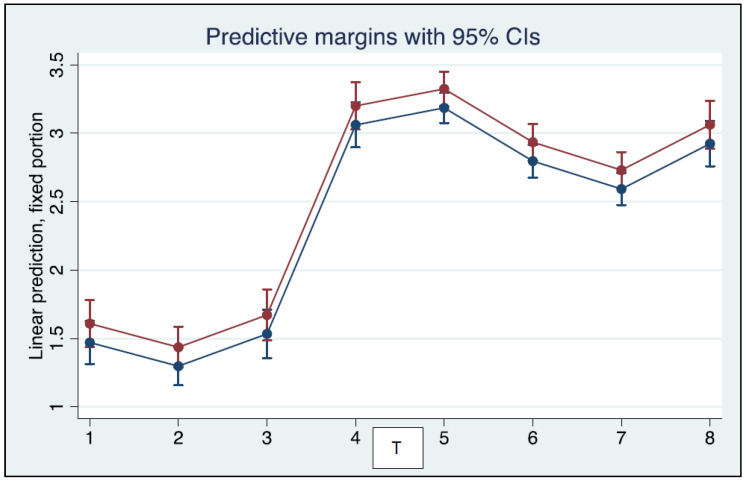
Log_10_ transformed neutralizing antibody levels as predictive margins after mixed-model linear regression. The effects of the previous symptomatic infection can be inferred by comparing the red line (symptomatic) to the blue line (asymptomatic).

**Table 1 vaccines-10-01237-t001:** Mixed-model linear regression, as mixed-effects ML regression. NtAb levels at T1 were the comparator levels. NtAb titer was expressed as ID_50_: the reciprocal value of the sample dilution that showed 50% protection of virus cytopathic effect.

log_10_ Antibody Level						
At Time point	Coefficient	Std. Err.	z	*p*	95% Conf. Interval	
T2	−0.1728	0.0929	−1.86	0.063	−0.3549	0.0093
T3	0.0632	0.0935	0.68	0.499	−0.1202	0.2465
T4	1.5908	0.0798	19.94	0.000	1.4345	1.7471
T5	1.7150	0.0843	20.34	0.000	1.5497	1.8802
T6	1.3259	0.0780	17.00	0.000	1.1730	1.4788
T7	1.1215	0.0756	14.84	0.000	0.9733	1.2696
T8	1.4534	0.0963	15.09	0.000	1.2647	1.6421
other variables						
female	−0.0388	0.0698	−0.56	0.579	−0.1756	0.0981
age	−0.0076	0.0036	−2.14	0.033	−0.0146	−0.0006
Symptomatic infection	0.1672	0.0679	2.46	0.014	0.0342	0.3002
intercept	1.8340	0.1739	10.54	0.000	1.4931	2.1749

Number of observations = 308, number of groups = 48, log likelihood = -78.54, Wald chi2(10) = 1655.99, *p* = 0.0000, coefficient = additive contribution of each explanatory variable to the value of the dependent variable, Std Err = standard error, z = the ratio between the regression coefficient of each explanatory variable and the related standard error. The symptoms of the infection were described by the binary variable “symptoms” as absent (0) or mild (1).

## Data Availability

The raw data can be made available by the corresponding author upon reasonable request.

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
