# Peer review of "Long-Term Longitudinal Analysis of Neutralizing Antibody Response to Three Vaccine Doses in a Real-Life Setting of Previously SARS-CoV-2 Infected Healthcare Workers: A Model for Predicting Response to Further Vaccine Doses"

_vaccines, 2022, doi:10.3390/vaccines10081237_

Round 1

Reviewer 1 Report

Despite the small sample size, this is an interesting study due to the multiple data point and long-term follow up of this particular population. The results are informative and useful to define vaccination strategy for healthcare workers. I suggest expanding the discussion and further elaborating on your conclusions/recommendations, given the current scenario of emergence of new variants.  

Strengthens:

· Well-written article, with clear description/presentation

· Long-term follow-up after 3 doses Covid 19 vaccination

· Many time-points testing of neutralizing antibodies

· Particular study population of previous mild or asymptomatic Covid 19 infection

· Closer follow-up to detect reinfection during 2 years period.

· Evidence of a booster immune response after 3rd dose

· Important evidence that can guide vaccination strategy for t healthcare workers

Weaknesses or area for improvement

· Small sample size

· It would be desirable to propose concrete new areas of research taking into account the dynamic of virus variants,  response to vaccination, and new variant-specific vaccines.

Reviewer 2 Report

Summary of the Work

The aim of this work is to provide a model able to predict the neutralising antibody response to further vaccine doses. The duration of protective immunity in an ongoing outbreak is determined by understanding the interconnection between infection and vaccination. To this end the authors studied the long-term dynamics of NtAb titers in healthcare workers with asymptomatic or mild wild type SARS-CoV-2 infection followed by a three-dose schedule of BNT162b2 mRNA vaccine and no reinfection up to more than 600 days.

Main Results Obtained

- The highest NtAb increase was following the first vaccine dose while subsequent doses did not further boost NtAb titers.

- The third vaccine dose appeared to revive waning immunity.

General Considerations

- Not all the acronyms have been specified. Please, specify them (even though they are well known in the literature) when they appear for the first time in the text.

- The topic, in itself, is interesting. However, the work is presented too concisely. In particular, the statistical analysis and the random coefficient mixed model analysis of variance (ANOVA) needs to be described in more detail.

- Some parameters introduced in the manuscript have not been defined.

- Section 5. provides a general recommendation rather than concluding remarks and/or perspectives for future works.

- It could be argued that the main result obtained by the authors (see above) is not innovative as it is exactly what we expected to obtain from the administration of the three doses of the vaccine.

- Written in this concise way, the work is vulnerable in several respects. The following tips are aimed to help the authors to fill some gaps.

Suggestions

S1) One of the main results found by the authors is briefly reported above. We may object that this was precisely the purpose of the three vaccine-doses i.e.,

- the scope of the first vaccine is to stimulate an NtAb increase;

- the second dose is intended to reinforce the effect of the first vaccine dose (without necessarily having to booster NtAb titers);

- the scope of the third vaccine-dose is to revive waning immunity.

However, the problem that are currently facing the research institutes is not "how long does the vaccine last?" but rather "is the current vaccine equally effective against SARS-CoV 2 variants?". The authors are asked to provide their comment about this generic question.

S2) Please, specify all the acronyms and key words introduced in the manuscript. For instance,

S2a) Figure 2 shows the antibody levels (in Log10) at the 8 time points. Please, specify the meaning of “minimum” and “maximum” and IQR.

S2b) In table 1, please specify “z” (is this the “test value”?) and “(capital) P” (is it the same as “p-value”?). In Table 1. what kind of Coefficients do the second column show? Are they referring to the Pearson correlation coefficients? If yes, when applied to a population, Pearson's correlation coefficient refers to a pair of random variables (X,Y). In this case, please, specify X and Y.

S2c) if X = Log10 transformed neutralizing antibody levels, please explain why this random variable is only related to one variable (Y in your case) and not to a series of random variables.

S3a) The aim of the statistical analysis is to establish a correlation between observations and the live virus NtAb titers. In Section 3., the authors presented their results by showing the p-values. However, we know that correlations are typically written with two key numbers: r and p. The closer r is to zero, the weaker the linear relationship. Positive r values indicate a positive correlation. Please, specify also the values of r.

S3b) The authors found that the third vaccine dose significantly revived waning NtAb 150 titers, by establishing a p-value inferior that 0.001. For the sake of clarity, the authors are asked to specify the null hypothesis in their case.

S4) Table 1 shows the standard errors and the confidence intervals related to the statistics for NtAb titers at Tj with j=2, ... ,8 (with T1 chosen as the comparator level). Finally, which kind of distribution has been found by the authors? Are the residuals normally distributed? Please specify by providing the interpretation of the obtained result.

S5) We come now to a crucial point. This study has been carried out by using the mixed model ANOVA. The authors on page 5, line 159 wrote: “Mixed model ANOVA, as mixed-effects ML regression”. However, there are several differences between the repeated measures ANOVA and the linear mixed models. So, written in this form, this sentence may induce confusion. Please clarify.

S6) As rightly done by the authors, the symptoms of the infection are described by the authors by the binary variable “sympt” as absent (0) or mild (1). However, as known, in this case, the most appropriate model is the generalized linear mixed models that work also for this type of dependent variables (generally, ANOVA is not considered as an option). This may be source of objection. Please, clarify this point.

S7) As mentioned by the authors, the main limitations of their method "are the low sample size, the incomplete availability of samples at intermediate time points and no evaluation of the cross-protection against the emerging variants". However, it is known that, under these limitations, repeated measures ANOVA can only use listwise deletion, which can cause bias and reduce power substantially. So, the use of repeated measures is suitable only when missing data is minimal. On the contrary, mixed models do a much better job of handling missing data.

This aspect constitutes one of the most vulnerable points of this work. The authors are asked to dispel this possible criticism.

S8) Another key point that may be source of objection is the following. This work requires a repeated measure over time, but subjects are also clustered in some other grouping: gender patients, patients having with age in a given time-interval, etc. Even in this case, it is commonly agreed that a repeated measures ANOVA cannot incorporate this extra clustering of subjects in some other clustering, but mixed models can. Hence, the natural question is: "why did the authors decide to use the ANOVA model instead of the more appropriate mixed model?".

Conclusions

Reading this work, I had the impression that the authors had written the manuscript in a very (I would say, too) hasty way. The authors should make an extra effort and explain with more care and in more detail the methodology they used and, above all, how they performed the statistical analysis of the data. Simply producing a short sketch is not useful, on the contrary, it leaves deeply unsatisfied the reader interested in continuing research in this field and in using the same authors' approach. Furthermore, the use of the ANOVA model has not been fully justified. I know, the mixed models, even if more flexible, they are more complicated with respect to the much simpler and easier-to-understand repeated measures ANOVA. Finally, I encourage the authors to take into account the suggestions expressed above. As said, these suggestions are only aimed to help the authors to fill some gaps.

Round 2

Reviewer 2 Report

The authors answered all the questions raised in my previous report. Some answers are satisfactory at others much less and, I believe, objections may arise on the part of the reader. For instance, to question 10), the authors replied: "The distribution of the regression coefficient assumed by the model is the normal, or Gaussian. When you see the" z "values, it is Gaussian". Fine, less obvious is however the value of the Kurtosis of this normal distribution. Furthermore, if the authors found that is a "perfect" gaussian, in this case it is legitimate to think that this profile is due to the "Central Limit Theorem". Indeed, we know that when many independent random variables are summed up, their properly normalised sum tends toward a normal distribution even if the original variables themselves are not normally distributed. In the specific case examined by the authors I do not see many random variables or, if the authors have focused their attention only on one variable, it is not clear why this variable is normally distributed. There were the contents of my question 10).

As for question 12), the authors are asked to briefly clarify what the mixed ANOVA model is, distinguishing three cases: 

a) ANOVA model,

b) Mixed model;

c) Mixed model ANOVA.

Otherwise, the reader risks getting lost (for completeness, please cite some relevant references related to the three above-mentioned cases)

Finally, as regards questions 13) and 14), the authors simply replied: "There is no conflict between the two models". This can raise several objections. The problem is that in this case it is commonly agreed that the ANOVA model is not considered the most appropriate model.

In conclusion, the contents of the work is good; less good is, however, the statistical analysis, or rather, the description on how the statistical analysis was carried out. I have the impression that the authors are in a hurry to publish the work, considering the description on how the statistical analysis was carried out as secondary.

I therefore ask the authors to make a further effort by answering with more precision to questions 10), 12), 13) and 14).

Round 3

Reviewer 2 Report

These two rounds of evaluation substantially confirmed the following:

1) The work is interesting and the authors have well motivated the objectives and innovative aspects of this study;

2) Some aspects relating to statistical analysis are not entirely clear to me, in particular on which statistical assumptions they based their calculations.

Concerning the nature of the random variable, the authors cited the Gelman and Hill sentence "the assumption of normality of the errors is of little importance in the validation of the regression". So, the authors simply stated that they do not recommend diagnostics of the normality of regression residuals. The authors adopted the method of regression predictions. However, we know that the regression prediction is based on following assumptions: i) linear relationship, ii) multivariate normality, iii) little multicollinearity and iv) no autocorrelation. Furthermore, this method is based on the validity of a strong assumption: v) homogeneity of variances (i.e., similar variances in different groups being compared).  s based on following assumptions: 1) linear relationship, 2) multivariate normality, 3) little multicollinearity and 4) no autocorrelation. Furthermore, this method is based on the validity of a strong assumption: 5) similar variances in different groups being compared. Nowhere have the authors discussed or justified the assumption of the aforementioned hypotheses. This leaves the reader unsatisfied.

By the way, concerning the effects of boosters over time - a key objective of this work  as recently published by the America Academy of Sciences, we know that T lymphocytes remain stable after contracting SarsCov2 for a period ranging from three to 21 months from infection. This important result should be taken into account when a model for predicting response to further vaccine doses is proposed. 

In conclusion, I propose that the authors justify at a medical level, at least qualitatively, the aforementioned assumptions underpinning their method based on regression predictions. In my opinion, this additional effort will be useful to avoid possible objections on the part of the reader.